# Body, Soul, and Spirit: An Explorative Qualitative Study of Anthroposophic Meditation and Spiritual Practice

## Terje Sparby

Department of Psychology and Psychotherapy, Witten/Herdecke University, 58455 Witten, Germany; terje.sparby@uni-wh.de

**Abstract:** This article presents the results of a qualitative study of Anthroposophic meditation, which arose in the German-speaking world in the early 20th Century focusing on cognition, self-development, and pro-social action. The objective was to explore this previously unstudied form of meditation. The current sample (N = 30) consists of long-term practitioners of Anthroposophic meditation. Semi-structured interviews, focusing on demographics, background, and phenomenology and interpretation, were conducted with these practitioners. The material gathered was investigated using thematic analysis. Seven main themes were found: Self, cognition, perception, affect, sleep, embodiment, and environment, and, among these, 32 subthemes. Potential avenues for further research are outlined. Some of these overlap with current approaches to meditation while others represent new areas of inquiry: Personal development with a focus on strengthening the self, introspection or contemplative inquiry, sensed presences, the experience of phenomenological atmospheres, consciousness in the sleep state, embodied aspects of meditation experience, the relationship between practice and daily life, and meditation challenges.

**Keywords:** anthroposophy; meditation; Rudolf Steiner

---

## 1. Introduction

This article explores the range of experiences of a group of practitioners of Anthroposophic meditation. Rudolf Steiner's work and the spiritual movement founded by him, Anthroposophy, are increasingly attracting more research interest. A recent milestone has been has been the inauguration of a critical edition of Steiner's work (Steiner 2013). There have been a number of publications appearing in philosophy and psychology over the last couple of decades (Schickler 2005; Sijmons 2008; Sparby 2013; Weger and Edelhäuser 2014). If we include Anthroposophic medicine (Kienle et al. 2013; Kienle et al. 2011; Hamre et al. 2014; Scheffer et al. 2012; Hamre et al. 2007) and Waldorf/Steiner pedagogy (Uhrmacher 1995; Sobo 2015; Oberski et al. 2007; Sobo 2014), which grew out of and are still part of the Anthroposophic movement, the field of Anthroposophic research is indeed relatively large. Anthroposophic meditation, however, has hardly received any attention. This led to the conception of the research project that this article is based on, which aimed to be an initial exploration of Anthroposophic meditation experience in general and among contemporary practitioners.

Anthroposophic meditation originated in the German speaking part of Europe during the early 20th century. Many influences can be pointed out, such as Christian mysticism, German idealism, Goetheanism, naturalism, Nietzschean/Stirnerian individualism, and the theosophical movement (Sparby 2016a). Structurally, Anthroposophic meditation can be seen to have the notion of human freedom, including, for example, intellectualism or a spirit of inquiry, anti-authoritarianism, and a scientific outlook, as a foundation. However, it also involves, perhaps paradoxically, an element

of religious devotion and surrender as a way of deepening the connection of the human being to its environment. Finally, Anthroposophic meditation aims at a union with the spiritual world that involves a strong emphasis on nature and culture; specifically, it aims at providing inspiration that can transform human society based on spiritual insight. Hence, transcendence and immanence are strongly connected. Ultimately, nature, human life, and the whole of reality are seen as inherently spiritual. In short, Anthroposophic meditation represents a kind of spiritual empiricism aiming at social reformation. To what extent the realization of this has been influenced by the spiritual practice of Anthroposophists during the last hundred years is hard to assess but, if we use socio-cultural influence as a measure, Anthroposophy must be considered to be one of the more successful subcultural spiritual movements of the 20th century. Not only has Anthroposophy had a strong impact on pedagogy, but also on medicine, art, literature, social banking, and politics. Main works by Steiner include *Wie Erlangt man Erkenntnisse höherer Welten?* (Steiner 1992), *Theosophy,* (Steiner 1995) and *Geheimwissenschaft im Umriss* (Steiner 1989). The secondary literature on Anthroposophic meditation is, however, somewhat scarce. Some authors have provided sourcebooks (Wehr 1983; Romero 2014; Zimmermann and Schmidt 2015) or attempted to deepen this form of meditation (Zajonc 2009; Schmidt 2010; Ben-Aharon 2016). In recent years, annual meetings by a network referred to as the Goethanum Meditation Initiative Worldwide has taken place, a network that seeks to connect Anthroposophic practitioners and develop this approach further.

When exploring Anthroposophic meditation using psychological methods, an inherent conflict may be noticed. Psychological research seeks to uncover how the human mind works while Anthroposophy itself already contains such a view. Anthroposophic meditation is a way of coming to know the deeper structure of the human mind and spirit and in particular to understand how it relates to the more fundamental, spiritual aspects of reality (Sparby 2017b). Anthroposophy contains claims that overlap, go beyond, and sometimes contradict typical contemporary views found in psychology, such as that consciousness is a exclusively a product of human physiology (Weger and Edelhäuser 2014). Hence, when a psychological researcher investigates Anthroposophic practitioners, both may be challenged. Indeed, as noted below, one participant of the current study did not want to participate because of doubts regarding whether contemporary psychological methods could adequately represent Anthroposophic meditation experiences. However, the spiritual or the Anthroposophic and the psychological perspective can be viewed as complementary. The contemporary psychological approach of developing theories based in data rather than philosophical ideas is very much in accordance with the spiritual empiricist attitude of Anthroposophy. In other words, Anthroposophy can potentially include the results of psychological studies and is not necessarily challenged by them. Only to the extent that psychology becomes reductionist, i.e., insofar as it denies the existence of the mind and spirit or sees them as epiphenomena, it can present a challenge to Anthroposophic practitioners. However, psychological research is not necessarily reductive. In this way, Anthroposophic and psychological methods and perspectives may be viewed as complementary.

Four analyses have been published from a larger research project on Anthroposophic meditation up until now, looking at the general structure of Anthroposophic meditation practice (Sparby 2017b), discussing methodological issues (Sparby 2016b), reporting on motivations for Anthroposophic meditation (Sparby and Ott 2018), and describing the effects of the Anthroposophic mantra practice and the so-called subsidiary exercises (Sparby 2018). These studies will be summarized in the following, providing an introduction to Anthroposophic meditation. The present article presents the main results of the qualitative analysis of interviews with Anthroposophic practitioners, consisting of seven main themes and 32 subthemes.

As found in a previous analysis from this, what motivates Anthroposophic practitioners can be represented by 14 themes ranging from external motivations (duty, self-regulation, and curiosity, etc.), internal motivations (developing higher capacities for knowledge, self-realization, self-improvement, etc.), and motivations of service (gaining knowledge for the sake of practice, service to the world and humanity, and realizing Anthroposophy) (Sparby and Ott 2018). The most common motivation among

the current sample (11 participants or 37%) was developing higher capacities of knowledge. A general developmental trajectory in relation to motivations may be one starting with external motivations, going through a phase of internal motivations, and ending with service motivations.

Furthermore, there are many different perceived effects that can be connected to Anthroposophic mantra practice, ranging from different cognitive, affective, and volitional. Some uncommon perceived effects are, for example, strengthening the self, cultivating a connection to subtle impressions, and sensing presences (Sparby 2018). The subsidiary exercises—a set of practices aiming at developing the capacity of thinking, willing and emotion in a virtuous manner—may also result in a host of different perceived effects. One overall perceived effect that is noteworthy is the stabilization of the practitioners. The subsidiary exercises, therefore, may represent a way of reducing the prevalence of the destabilizing or challenging effects of meditation.

Much further work needs to be done in order to better understand Anthroposophic meditation and the potential effects associated with it. The current study gives an overview over the range of perceived effects of Anthroposophic meditation in the current sample. Future avenues of study include investigating the quantitative aspects of practice, particularly looking into the prevalence of perceived effects and investigating the effects of Anthroposophic meditation in relation to established psychological measures. It is also important to find new ways of measuring the specific effects connected to Anthroposophic practice, and the present study represents a contribution to this endeavor.

It may be noted that the main title "Body, Soul, and Spirit" refer to a way the human being is typically conceived of in Anthroposophy (Steiner 1995). This conception of the human being will be picked up again in the discussion and the relationship between the Anthroposophic categories and the categories presented in this study will be indicated.

## 2. Method

This section presents the qualitative approach and research paradigm (Section 2.1), researcher characteristics and reflexivity (Section 2.2), context of the study (Section 2.3), sampling strategy (Section 2.4), ethical issues pertaining to human subjects (Section 2.5), data collection methods and data collection instruments and technologies (Section 2.6), units of study (Section 2.7), data processing (Section 2.8), and data analysis (Section 2.9).

### 2.1. Qualitative Approach and Research Paradigm

The range of experiences that arise as a result of Anthroposophic meditation have yet to be studied under rigorous employment of established scientific methods. A central question, therefore, is deciding which methods should be chosen for an initial investigation. While quantitative measures are appropriate for "areas of research where much is already known", qualitative methods function well "where little is known about the phenomena in question" (Velmans 2000, p. 181). In general, qualitative methods are typical of exploratory studies (Given 2008, p. 327) and "are used to investigate new areas of inquiry or previously unknown types of behaviors, groups, or contexts" (Tewksbury 2015, p. 210). Furthermore, qualitative methods are employed "to explore the human elements of a given topic, where specific methods are used to examine how individuals see and experience the world" and "to explore new phenomena and to capture individuals' thoughts, feelings, or interpretations of meaning and process." (Given 2008).

Anthroposophic meditation represents a new area of inquiry, a previously uninvestigated form and context of meditation. Qualitative methods excel when it comes to gathering initial descriptions of the different individual experiences and phenomena associated with this practice.

The research design of this study was inspired by the methodology of the Varieties of Contemplative Experience project (VCE) (Lindahl et al. 2017) and some initial training was acquired from members of the VCE research team of Brown University. While the VCE project focuses specifically on challenging meditation experiences, an area hardly investigated, this project seeks to uncover the entire range of Anthroposophic meditation experiences, regardless of whether they are positive or negative. There are

presently no descriptions of Anthroposophic meditation experiences that can be found in the literature. The interviews and analyses were conducted by the Principal investigator while receiving feedback from the research group at the University of Giessen, whose other team members are working on a replication of the VCE study.

### 2.2. Researcher Characteristics and Reflexivity

The P.I. was a postdoctoral researcher at the time the study was conducted and had previous training and experience in conducting qualitative interviews. All interviews and analyses of the transcripts were conducted by the P.I. Some of the interviewees were his friends and/or colleagues and he has engaged extensively both with Anthroposophic theory and meditation practice in addition to other meditation practices (primarily Buddhist). This may lead to the objection that the researcher is biased, particularly if one takes a positivist stance. From a post-positivist or constructivist stance it may be countered that "unbiased" research is a chimera. Everyone will come to a research object with specific interests and perspectives that will be shaped based on previous experience and reflection. Rather than attempting to isolate such factors, it is better to make them explicit and include them as part of the research process. Making one's interests and perspectives explicit can also be viewed as a way of achieving a higher degree of objectivity. Such a process of explication is vital in order to potentially bracket one's prejudice (Giorgi 2009). In other words, positivist and post-positivist stances are not necessarily mutually exclusive. Reflecting on one's own presuppositions, subjective interests and perspectives can be a means of becoming aware of and sensitive to other points of view, potentially giving rise to new views that are more comprehensive. The approach taken here is to strive towards objectivity and wholeness by both continually reflecting on contextual factors and potential sources of prejudice while making these explicit when deemed relevant (such as when the researcher's background may influence an interpretation).

### 2.3. Context of the Study

The participants were all of European origin and the interviews were conducted in Germany, Switzerland, and Norway. With a few exceptions, the setting was a secluded and quiet space, often in the participant's home. An important contextual factor is that talking about and discussing one's own meditation experience with others has up until recently been problematic within the Anthroposophic community. The reasons for this include that it is looked upon as potentially detrimental to one's meditative development and that it is a personal and intimate matter. This was directly reflected in some interviews where participants communicated that they were reluctant about describing certain significant experiences. In some cases, they went on to describe those experiences; in others, they chose to not disclose anything further. In one case, a participant first chose not to describe a certain aspect of her experience and then changed her mind later in the interview. A further aspect of this is the potential tension between the original works of Steiner and new content presented practitioners of his meditative method. Steiner's texts enjoy widespread approval within the Anthroposophic community. His lectures and books are still regarded as representing deep spiritual insights. Presenting results from one's own meditative practice may be viewed as a challenge to the original texts; anyone doing so risks being met with disapproval from at least some parts of the community, as evidenced by some of the participants in this study. This is one likely reason for why it has been very uncommon to talk about and discuss one's own meditative experience within the Anthroposophic community.

In the recent decades, however, it has become more common and less problematic to speak openly about one's meditative practice and experiences (Sparby 2018). The interviews that this study is based on have revealed many examples of how this process has unfolded. This study can itself been seen as part of the process towards openness, although the fact that the anonymity of the participants is retained and contextualized within a Non-Anthroposophic framework makes participation less of a potential social risk.

### 2.4. Sampling Strategy

The participants were recruited using a combination of purposive sampling (Tashakkori and Teddlie 2003), snowball sampling (Faugier and Sargeant 1997), and special case sampling (Teddlie and Yu 2007). Since this was going to be the first qualitative study of the range of experiences relating to Anthroposophic meditation, the general aim was to recruit experienced long-time practitioners who can potentially report on a wider range of topics and phenomena than inexperienced ones. The P.I. was already acquainted with some experienced practitioners. Further participants were requited based on the recommendation of those who had already participated (snowball sampling). Since difficult meditation experiences may be underreported, cases representing prominent difficult experiences were specifically sought after when asking the participants whether they knew of any. Certain basic inclusion criteria were also adhered to: The participants had to be at least 18 years old, Anthroposophic meditation should be a main element of their meditation practice, and they should have experienced at least one significant or unexpected effect, whether positive or negative, as a result of their practice.

Although gender balance was strived after, it was not possible to achieve. This was in part due to the limited timeframe of the project and the resources available to complete it, but also due to the higher rate of hesitancy among female candidates. All potential female participants suggested by the other participants were contacted, but this was not enough to attain gender balance.

### 2.5. Ethical Issues Pertaining to Human Subjects

Prior to contacting any participants, the design of the study was reviewed by the local Institutional review board at the University of Giessen. Potential participants received a document outlining the nature and purpose of the study. The document also contained information about potential risks that participation could entail. The participants were given a unique ID number, but the transcripts were anonymized by deleting the name of the participants, names of family members or significant others, place of birth, universities attended, and current residency.

### 2.6. Data Collection Methods and Data Collection Instruments and Technologies

The interviews were conducted between December 2014 and July 2015. All interviews were conducted by the P.I. in-person, except for two that were done via real-time electronic video and/or audio transfer. All interviews were audio recorded. The average duration of the recordings was one hour and 41 min (the shortest interview being 49 min and the longest 2 h and 23 min). The interviews followed the same format, starting with demographic questions and proceeding with an open structure. The interview protocol is presented in the Supplementary Materials S1.

### 2.7. Units of Study

Thirty subjects participated in the study (see Table 1 for demographic characteristics). Five potential candidates declined to participate, three did not respond to the invitation, and three were, for various reasons, hesitant about taking part in the study and were, in the end, not interviewed. Hence, 11 potential candidates either declined to participate, did not respond, or did not participate due to hesitancy. Of the candidates that declined, one did not have enough time available, one had doubts about the validity of academic research on meditation experiences, and one declined because obtaining consent was perceived as being too bureaucratic. Two declined for unspecified reasons. One potential participant that could possibly have represented a case of strong difficulties connected to her meditation practice did not respond. One of the hesitant candidates expressed doubts about describing her meditation experiences because, as it was stated, it could make those experiences disappear, and another doubted that her experience could be counted as significant enough to be included in the study.

**Table 1.** Participant Demographics. Numbers indicate prevalence in the present sample (N = 30) except in the case where years are stated.

| Demographic Characteristics | |
| --- | --- |
| Female (%) | 9 (30%) |
| Male (%) | 21 (70%) |
| Mean age (years) | 51 |
| Average duration of regular practice (years) | ca. 21 (range: 1.5–41) |
| European nationalities (%) | German: 14 (47%), Norwegian: 6 (20%), Swiss: 3 (10%), British: 2 (7%), Belgian: 1 (3%), French: 1 (3%), Austrian: 1 (3%), Russian: 1 (3%), Danish: 1 (3%) |
| Spiritual orientation of parents (%) | At least one parent: Anthroposophic: 6 (20%) Catholic: 9 (30%) Evangelical: 13 (43%) Agnostic, atheist or non-aligned: 8 (27%) |

*2.8. Data Processing*

The recordings were transcribed primarily by the P.I. All participants were offered to read through, expand and/or correct the transcripts. Three participants chose to do so, but the changes to the original transcripts were minimal.

*2.9. Data Analysis*

Since this study is exploratory in nature, no pre-determined definitions of themes were used. The transcripts were first analyzed using software (MAXQDA) following open coding techniques (Strauss and Corbin 1998) as part of an overarching thematic analysis approach (Guest et al. 2012). The aim was to let themes emerge from the text (inductive or data-driven coding). In the further analysis, some theoretical constructs have been used, such as when analyzing the motivations for Anthroposophic practice, resulting in a combination of inductive and deductive strategies, which can both be part of thematic analysis. The first draft of the codebook was created by the P.I. alone and the results were presented to the research group at the University of Giessen. Then, after the initial feedback, rough definitions of themes were refined with clear inclusion and exclusion criteria for each theme. The codebook was then validated by an external researcher. This process consisted of six interviews being coded using the existing codebook. The identified themes were then compared to each other. In particular, themes that could have been missed were looked for. However, this resulted in only one additional code (the affective category of "sadness"). After this, each interview was re-read and the identification of the themes re-checked. This resulted in very few revisions and the codebook was finalized. The codebook is contained in Supplementary Materials 2.

**3. Results**

In the following an overview of practices will be presented (Section 3.1) as well as the themes that were discovered by the qualitative analysis (Section 3.2).

*3.1. Overview of Practices*

The most common practices are presented in Table 2. Anthroposophic practices are considered to practices that have been suggested originally by Steiner or common practices within the Anthroposophic community growing out of Steiner's work (for example biographical work), while practices that come from other traditions, such as Christianity or Buddhism, are considered to be non-Anthroposophic practices. The practice and effects of the most common practices, mantras and the subsidiary exercised ("Nebenübungen") have been described elsewhere (Sparby 2018).

**Table 2.** Overview of Practices. Numbers indicate prevalence in the present sample (N = 30).

| Overview of Practices | |
| --- | --- |
| Most popular main practices: Number of participants/(%) | Mantras or "Sprüche": 26 (87%) <br> Mantras of the school of spiritual science. 14 (47%) <br> Foundation Stone Meditation ("Grundstein"): 10 (33%) <br> "He thinks, she feels, he wills": 9 (30%) <br> Subsidiary practices ("Nebenübungen): 25 (80%) <br> The rosy cross meditation: 14 (47%) <br> The daily review (Rückschau): 14 (47%) |
| Other Anthroposophic meditations represented | Vocational meditations: 1 (3%) <br> Thought meditations: 10 (33%) <br> Practices for the subtle body: 5 (17%) <br> Meditation on feelings: 2 (7%) <br> Karmic meditations: 11 (37%) <br> Cosmological meditations: 3 (10%) <br> Empty consciousness: 8 (27%) <br> Visualizations: 17 (57%) <br> Perceptual meditations: 17 (57%) <br> Meditations on beings: 5 (17%) |
| Other traditional Anthroposophic practices | Study 8 (27%) <br> Biographical work 1 (3%) <br> The calendar of the soul/poems: 3 (10%) <br> Projective geometry: 1 (3%) <br> Asking for help before falling asleep: 1 (3%) <br> Eurythmy: 6 (20%) <br> Developing interest: 1 (3%) <br> Speech formation: 1 (3%) |
| Most popular other forms of meditation or spiritual practice (%) | Prayer: 10 (34%) <br> Mindfulness meditation: 8 (27%) <br> Open monitoring: 7 (23%) |
| Practitioners reporting meditating up to or more than one hour a day (%) | 6 (20%) |
| Practitioners reporting meditating more than two hours a day (%) | 0% |

*3.2. Themes*

Seven main themes were identified by the qualitative analysis: Self, Cognition, Perception, Affect, Sleep, Embodiment and Environment. These themes and their subcategories will be described below. Note that the words "theme", "experience" and "effect" are sometimes used interchangeably and that that when there is a reference to effects of Anthroposophic practice reported by the participants of this study it is always a matter of a "perceived effect", although this is not always specified. A taxonomy of the experiences is presented in Table 3. Note that "experience" is a quite broad category and includes in principle anything the participants report on having gone through as a result of their Anthroposophic meditation practice; some of these experiences may last only a few seconds or even less, while others are transformative experiences taking place over years.

Note that all themes may contain both positive and negative or challenging experiences. Eleven (36%) of the participants mention challenging effects lasting for more than one day in connection to their Anthroposophic meditation practice. The challenging effects range from minor challenging effects such as feeling imbalanced, isolated, or that life becomes more difficult, to effects that may be connected to psychological disorders: Depression, hearing voices, phobia, psychosis, anxiety, sleeping disorder, and hypersensitivity. However, with only one exception, none of the challenging effects resulted in a functional impairment during daily life. The exception concerns a participant who experienced

a burnout. Meditation was believed to be one potential factors leading to the burnout, but not the only one.

**Table 3.** Overview of Themes and Prevalence of Themes in the study sample (N = 30).

| Main Theme | Sub-Theme (Number of Participants Reporting This Theme/Percentage) |
|---|---|
| *3.2.1. Self* | Character (26 (87%)) <br> Self-encounter, Self-Strengthening, and Self-Dissolution (26 (87%)) <br> Capacities (17 (57%)) <br> Crisis (17 (57%)) <br> Oneness (9 (30%)) |
| *3.2.2. Cognition* | Knowledge (24 (80%)) <br> Thinking (13 (43%)) <br> Memory (8 (27%)) <br> Conviction (6 (20%)) <br> Attention and Meta-Cognition (4 (13%)) |
| *3.2.3. Perception* | Sensed Presences (28 (93%)) <br> Visual (23 (77%)) <br> Sensations (23 (77%)) <br> Auditory (18 (60%)) <br> Taste and Smell (5 (17%)) <br> Touch (5 (17%)) |
| *3.2.4. Affect* | Joy (16 (53%)) <br> Fear (15 (50%)) <br> Peace and Calmness (8 (27%)) <br> Sadness and Depression (7 (23%)) |
| *3.2.5. Sleep* | Dreams and Lucid Dreams (11 (37%)) <br> Sleep Paralysis (6 (20%)) <br> Lucid Dreamless Sleep (5 (17%)) |
| *3.2.6. Embodiment* | Vitality and Health (14 (47%)) <br> Grounding (13 (43%)) <br> Energies/Forces (14 (47%)) <br> Energetic Centers (11 (37%)) <br> Out of Body Experiences (14 (47%)) |
| *3.2.7. Environment* | Relationships (19 (63%)) <br> Life Competency (13 (43%)) <br> Integration (11 (37%)) <br> Meaningful Connections (4 (13%)) |

### 3.2.1. Self

The theme "self" contains all experiences and effects of meditation practice that relate to the self. This theme includes motivations and in particular motivational changes that arise due to meditation practice. Motivations and motivational changes have been described in-depth elsewhere (Sparby and Ott 2018); what has been shown is that there are three main forms of motivations—external, internal and service—and that the interviews tend to show a development from the former to the latter.

Some of the themes below draw a connection to the traditional Anthroposophical terms imagination, inspiration, and intuition. These can be understood to correspond to visual, auditory, and unitive experiences happening in altered states of consciousness arising through meditation. Steiner developed these terms mainly in *Stufen der höheren Welten* (Steiner 1993). The terms are quite complex and can be used in a variety of different senses. For example, although "imagination" mostly refers to visual experiences in altered states of consciousness, they can also refer to seeing figures within everyday consciousness (Sparby 2017a). Furthermore, imagination as an altered state of consciousness contain many different aspects, such as light, colors, radiance, inherent intelligence, etc. (Sparby 2020).

It is not always clear which aspects the reports concern, and in the following the emphasis is on the experience itself rather than the Anthroposophic interpretation of the term.

Character

Some meditation experiences concern changes or an increase/decrease in character related properties such as virtues and vices. These include further subthemes such as self-determination, autonomy, and selflessness, but also egotism, uncertainty, and self-accusation. Some participants, for example, described an increase in arrogance connected to their practice, but also that meditation became a source of moral action and an ethical deliberation. For them, meditation enabled a connection with the forces of good and the will to serve, to be there for something other than oneself.

Self-Encounter, Self-Strengthening, and Self-Dissolution

A spiritual crisis sometimes led to discovering more fundamental aspects of the self, but such self-encounters could also happen independently of a crisis. Some participants described encountering a "higher self", initiatory experiences, or that the self was strengthened, which typically means that the sense of personal identity over time increased or came into alignment with one's true life intentions. Some experiences could be quite simple. One participant stated that meditation can lead to an increased sense of having a second home. For further and more specific participant descriptions, see the article (Sparby 2018). It is also possible that the self dissolves as part of the meditative practice. As one participant described it, "it is a kind of death, a death experience. Nothing is left of the [name of participant] that I knew [ . . . ]. The self that I had known in that way so far, I cannot take with me". Note that the idea of the higher self in Anthroposophy distinguishes itself from other traditions such as Buddhism, where the existence of the self is mostly denied. Although there are examples of true self doctrines for example in Mahayana Buddhism and in other Indian traditions, these doctrines present the true/higher self as a universal self. In Anthroposophy, the higher or true self is understood as having an individual aspect, so that someone who experiences an ego dissolution can still find a true identity, a true home, that is also personal.

Though an ego dissolution could be a positive experience on a whole, especially if a new identity arose, it could also be depersonalizing. However, only two cases that could be described as depersonalization were found. As one participant stated:

> "I was meditating a lot, and reading a lot, and I overdid it basically. And I had an experience where I sort of suddenly lost a sense of self. Which was very scary actually. And I really couldn't get a grasp, suddenly, of who or where I was. And it was literally like, you know, the self just kind of disappeared for a while, and I had to . . . I just had to . . . I just . . . luckily, I was at home. I just did the most familiar thing that I could think of doing, kind of, and I just had to keep active doing something very simple, and familiar. And gradually, kind of, normalized, and I was ok again, but it was very, very uncomfortable."

Furthermore, one participant reported on the possibility of becoming "dreamy", which consisted of a lack of connection to everyday reality and an inflation of the ego, which in the context of this theme can be interpreted as a dissolution of the everyday self, accompanied by a strengthening of a pathological (disconnected, grandiose) self.

Capacities

Different capacities were perceived to arise from the Anthroposophic meditation practice. As a participant noted, emotions or moods such as devotion to truth could be evoked within in a few of minutes. Through meditation, one could also experience an increase in one's capacity for self-regulation. Some participants described acquiring the ability of inducing an out of body experience, which was sometimes understood as connecting the practitioner with the spiritual world (see discussion in (Sparby 2018)). Other participants described other capacities, such as new perceptual abilities, becoming



more able to sense certain fine/subtle sensations or atmospheres and even presences (see Section 3.2.3). Specific practices, such as the subsidiary exercises, were connected to the development of specific capacities as well, for instance the ability to maintain a series of thoughts on the same subject (see list in (Sparby 2018)). Finally, though descriptions relating to the feeling of love were rare, one significant experience in relation to a sensed evil presence (a more in-depth description is presented below) led to an understanding that, through being able to embrace and transform the evil presence, a greater capacity for love and a tolerance for the evil actions of others would develop. Furthermore, another participant noted that the capacity for love deepened through meditation practice.

Crisis

Participants reported undergoing biographical or spiritual crises as part of their meditative development. This included encountering repressed and underdeveloped aspects of the personality. Although crises could lead to breakthroughs in personal development, they could also be associated with periods of inactivity and reduced functionality in daily life. Though meditation could be a factor in triggering a crisis, it could also provide the practitioner with support. Sometimes a crisis was described as a "threshold experience", which refers to liminal state where the old is being left behind, but the new has yet to arrive; one of the general aims of Anthroposophic practice is to come to know a spiritual reality through direct experience, which can include a radical break with one's previous identity or reality.

Oneness

The Anthroposophic meditators of the present sample occasionally reported on experiences where the self was in some sense one with something external to it. Examples included experiences of oneness with nature or the periphery, but also experiences of transcendence or enlightenment. As one participant described it:

"It is not something I strive to attain, but still it can be delightful, although it is rare. [ . . . ] [I] enter a state, I don't want to say out of body, but I am . . . I could meditate for hours. It is as if I am inside a glass, in a world that is living, I can perceive everything [ . . . ] I don't have to move an eyelash or anything else, it's an experience of oneness with everything, an enlightenment experience."

This experience is an example of a kind of oneness experience that is unspecific. Other experiences of oneness were specific, i.e., the self of the practitioner is experienced as one with a certain content, which can also be external to the practitioner, such as rocks or minerals.

3.2.2. Cognition

This theme relates to changes in cognitive processes and contains the following sub-themes: knowledge, thinking, memory, conviction, attention and meta-cognition.

Knowledge

About the theme of knowledge includes concrete insight gained through the meditative or spiritual practice. *Perception*, which is also knowledge related, is a separate theme (see below). The *knowledge* theme focuses on conceptual or discursive knowledge and intuitive insights. Anthroposophic meditators explicitly make use of what is sometimes referred to as contemplative inquiry, which typically consists of a combination of focus attention and open presences or empty consciousness meditations. One participant described working on topics such as the meaning of suffering and sacrifice, and stated that this kind of work often culminates in a meditative experience. The work could take years and was enriched by daily life experiences until everything came together in a meditative event. Such events could be lightning fast, bringing with them a plethora of different

ideas and associations. Some participants also described coming to an understanding of how the human being is constituted, such as what processes underlie everyday cognition. This could be called introspective knowledge. For example, one participant stated that he could retain the meaning of a word without the word itself: "there is a process of wordless understanding, which is more primordial than the conceptual". This wordless sphere can also give rise to insights that appear as specific ideas. A participant mentioned a criterion he used for whether such insights count as real insights: They have the quality of originality, or as he put it, "I see it like that for the first time", adding that the insight was accompanied by a sense of novelty and freshness.

Thinking

This theme relates to changes in the thought process and includes insight into how thoughts arise, the ability to think clearly, and the ability to stop the thought process at will. Note that this theme does not treat concrete knowledge beyond that which belongs to the thought process. Some Anthroposophic meditations work specifically on the thought process, and the effects are for instance insights, increased structure of thought, and a development of trust in thinking (see full list in (Sparby 2018)). One participant stated that it becomes easier to think through such practices, that the trust in thinking grows, and that he had insights into the nature of thoughts (for example, that their pure conceptual content is independent of material processes). Another participant described how the fact that thinking can often be difficult (some notions and thought processes are "chewy" or tough) may lead to an increased sense of oneself, although this does not preclude the possibility of giving other people or phenomena room in one's thinking.

Memory

The ability to remember can increase or decrease through meditation. Besides developing a better capacity for memory, an increase can also come in the form of de-repression, i.e., that old traumatic experiences appear in consciousness. One participant described a particularly strong event happening after a meditation session while he was lying down on a sofa:

> "I start to cry, and then a whole lot of memories come up, a whole lot of things that I remember from growing up when I hurt someone. I laid there and struggled terribly with pangs of conscience really. All the times I have hurt someone came up. From when I was quite small. Things I've ... if I had bullied a teacher at school, or bullied some boy at school, or a girl I had broken up with because I was in love with someone else. All of these experiences where I had hurt another human being. They came up."

This de-repression happened as part of an event that was a self-encounter in the sense described above and ended up being a deeply transformative experience for the practitioner, affecting, among other things, his career path.

Conviction

A consequence of insights can be that the practitioner gains a stronger conviction in the fruitfulness of their spiritual endeavors. One participant stated that they had gained such a strong conviction through their practice that they were as sure that there is a spiritual world as that there is a table that they can touch in front of them. This is a matter of "inner firmness" and gives rise to a belief or trust in the Anthroposophic project and movement as a whole, insofar one has confirmed a part of it for oneself. Experiencing that part of it for oneself gave grounds to believe that the rest of it was true as well. The converse of this is a lack of conviction or doubt that can arise if the practice leads to no specific results or insights. It may also be noted here that finding out for oneself whether the claims of Steiner and Anthroposophy are true was a motivation for meditation that most practitioners of this study had in common (Sparby and Ott 2018).

Attention and Meta-Cognition

Meditation practice can lead to an increase of attention and meta-cognition, i.e., the ability to be aware that one is attending to something and what one is attending to at a given time and potentially act in a beneficial way. Not only an increase of attentional focus in a simple sense (the ability of directing and sustaining attention on an object), but, as one participant noted, it also uncovers aspect of consciousness that one was previously unaware of, which can contain difficult psychological material:

> "And that is what I mean, that meditation . . . there are different directions. One is to become more grounded and so on. And meditation also helps to bring up things purposefully that lie just below consciousness by listening for it. I found that very helpful, that has helped me a lot, and also helps me in different situations. For instance when I am agitated or something. Then I notice that if I practice meditation, then that helps me notice more clearly where things are bubbling up, and also to see the relationships more clearly."

Furthermore, one participant pointed out that meditation can increase attention in the sense that one notices important events in daily life, which enables making the right decision or even making a decision at all. For example, it can be important for teachers to notice the need of a student in a particular situation. This is related to meta-cognition. As one participant noted, we are constantly having different ideas coming into our mind, i.e., attention shifts from different thoughts again and again. We need to know that we are having those thoughts to be able to retain them so that they can be made use of in relevant contexts. Meditation increased this ability of meta-cognition. Another attentional ability that was mentioned was the ability to empty consciousness completely, which is a kind of negative attentional ability. In empty consciousness, attention should have no content and Anthroposophic meditators view this as a condition particularly useful for enabling new content to appear in consciousness.

### 3.2.3. Perception

The theme of perception can be divided into aspects that have to do with quantity and quality. Many participants described an increased ability of perception or that one can become more sensitive and attentive to the percepts; this is the quantitative aspect. Some reports also indicated that it is possible to become too sensitive as a result of practice. One participant described becoming "vulnerable". The qualitative aspect contains perceptions associated with the commonly accepted senses: *Visual*, *auditory*, *olfactory*, and *touch*. Some participants spoke of *sensed presences*, but also changes in the perception of *space* and *time*. Some participants reported on synesthetic experiences, such as a seeing that is also a hearing or a touching that is also a seeing. This can be described as a change that is both qualitative and quantitative. A single percept contains more information than usual and has also changed quality in that two modalities have merged. In the following, only the qualitative aspects will be described further.

Sensed Presences

"Sensed presences" is one of the themes that contain the greatest amount of rich descriptions. One of basic phenomena seems to be a simple experience of presence. One participant compared it to what it is like to stand in front of an animal or seeing a fish in an aquarium. There are also descriptions of a sense of being watched. In Anthroposophy, there are many terms for describing such presences and are referred to using traditional names. Examples of sensed presences are: Beings of nature, plants, sensing nature as a whole, a Doppelgänger or shadow, angels, archangels, Christ, etc.

The interviews show that the experience and interpretation of a sensed presence can be distinguished. An "angel" could, for example, be described as consisting of an experience of light, goodness, and a localized sensed presence. That it is an angel is in the view of some participants, an interpretation that can be influenced by one's background and social environment. Encounters with sensed presences could be very significant, and both positively and negatively valenced. Sometimes

they were connected to experiences of fear, disgust and being attacked, but they could also involve a sense of being protected, receiving guidance, and new capacities.

Some of these experiences happened during sleep or when going to sleep. For instance, one participant described being strangled by a sensed presence during the night and also sensing presences in the background while going to sleep.

Visual

The visual aspect of meditation experiences is central to Anthroposophic practice and this is reflected in the richness of different visual reports. The most general visual experience is that of "light", but it also includes seeing images. The traditional term for visual experiences in Anthroposophy is "imagination" and the theme of visual experiences includes all those reports as well.

The descriptions of light experiences range from simple impressions to blinding and exploding light. Strong light experiences are, however, rare. On two occasions, strong light experiences were connected to illness and death. One participant described meditating in the evening, then probably falling asleep for a short moment, after which he wakes up and experiences his head being filled with light. He focused on the light for the rest of the meditation, lost track of time and meditated through the whole night. No similar reports were found but some participants reported experiencing light impressions during meditation and also in relation to other beings and even natural objects such as crystals. One participant, who reported having many such impressions, described the light as connected to happiness and a great, focused presence. Besides this, many different qualities are connected to light impressions: It could be different colors, such a silvery or blue or can be diamond-like, like a lightning strike streaming upwards and radiating.

Sometimes light experiences were connected to the appearance of images as well The nature of the images experienced was highly complex and it is not be possible to give full justification to the richness of the reports here. The intensity or clarity of the images in general could vary a lot. Some participants described visual images that were similar to sensory experiences or inner representations, like memories. Others described quasi-visual imagery, where for instance the affective component of a color rather than its visual aspect was perceived. Some participants mentioned seeing images, faces, figures, and lines. Some images contained an auditory or cognitive component as well (as if information were contained in the image), some were experienced as cold, and other images were experienced as healing or invigorating. Indeed, it seems that images often contained other secondary modalities like this.

Sensations

There are series of experiences that cannot be easily understood as relating to a specific cognitive function, sense, or affect directly, though they may be related to them indirectly. These experiences can be referred to as sensations. Sensations can be understood as a reaction *within* the meditator but also as corresponding to something *beyond* the meditator even though the sensations do not relate directly to any physical sense organ. These sensations are sometimes referred to in general as moods, but they also often relate to other aspects of the human being: "I sense moods of nature that do not come from the senses [ . . . ]. It is not yet concrete perceptions [ . . . ] but rather like a basic rustling behind that, behind nature." In this quote, moods are possibly understood as a form of proto perception—something that may develop into a perception that still only consists of a mood or subtle sensation.

Some sensations were more cognitive, such as wisdom, mercy, and dignity. Some had a more sensory dimension, such as radiance, freshness, dryness, and bitterness. Others were more affective, such as love, intimacy, and suffering. Others again had a more vital or bodily character, such as health, vitality, warmth and coldness. Finally, there were sensations that may relate to many different dimensions, such as beauty and clarity.

Auditory

Auditory experiences also come in a range of different varieties. Some participants spoke of hearing voices internally that have a quality similar to hearing someone speak (the voice can for instance be identified as either male, female or androgynous), while others stressed that it is only *like* hearing someone speak, but that no actual voices could be heard. Sometimes the voices involved negative self-talk, but they can also bring insights, or advice for practice or moral advice. Auditory experiences like these were often referred to by practitioners as inspirations, yet they often appeared together with visual experiences. Some insights or inspirations could come as wholes; one practitioner mentioned receiving a meditation consisting of words that were inwardly dictated to him. The experience of having insights was by one practitioner described as similar to how one remembers something one has forgotten. Another described it as something that is working its way into one's thinking. External auditory experiences were also possible. One participant, for example, described hearing the whispering of a sensed presence in nature. One practitioner stated having heard something that is like a song, another having heard something that is like "a giant organ of angel voices, ranging from deep, deep, deep, bass tones to tones that are so high that they can hardly be heard [ . . . ]. And it was very plastic, melodic, and harmonic." Finally, a loud noise was described as being connected to out-of-body events.

Taste and Smell

Only a few participants reported on taste and smell experiences. Some were connected to sensed presences. One participant for example described sensing the smell and taste of Christ. This experience was described as similar to a sensory experience, yet it had an emotional intensity and intimacy. Although it is reported to often appear on the right side, it was not within the mouth but rather "all around". Another participant described smelling something burnt right before a sensed presence of a more challenging nature appeared.

Touch

Sometimes touch was described in relation to initial perceptions that later grow to become more defined (involving other sensory dimensions) and touch experiences could be related to more synesthetic experiences. For instance, one participant spoke of "a tactile seeing" when referring to spiritual perceptions, and another spoke of feeling a form of touching. One participant reported on something he called a "spiritual touch", which he understood as an intensification of the capacity of thinking, similar to training a muscle. Participants also used the word touch in the sense of attentional scanning. Finally, when describing how he perceived sensed presences, one participant stated: "Methodically it is like I would stretch out my capacity for empathy, my feeling capacity, and touch the area; and then there is a felt resonance, there is a presence there".

Time Perception

A distortion of the perception of time was already indicated above (losing track of time). Another participant described a sense of timelessness arising in deep meditation. Something could be experienced as timeless although in reality only about half a minute or so would have passed. Judging from the internal sense of time, an hour might as well have passed. This experience was also connected to happiness and peace.

3.2.4. Affect

A large range of human affects were described during the interviews. The five most commonly experienced affects have been selected. Other affects were typically not described in-depth and hence the descriptions were not very informative, nor could specific patterns of experience be discerned.

Joy

Some participants mentioned the practice of meditation itself was connected to joy. In the words of one participant: "When I meditate, what most often gives a sense of calm, of being nurtured, yes, somehow receiving spiritual nutrition, and of this simply does one good, with [ … ], yes, with a feeling of joy". And although joy could be experienced in meditation, it could also be experienced in daily life when aspects of practice have an influence on it. One strong experience of joy happened in connection to realizing one's life purpose, and such realizations often happened in relation to having an insight, an experiential breakthrough, or sensing that one is accessing a greater spiritual reality. Though some experiences of joy come as a result of having an experience, others were connected to certain phenomena: Some participants described sensations of light as joyful, while others described joy in relation to sensed presences, such as experiencing Christ or the being of a flower. The joy experienced can be one's own joy, but can also be perceived as being the joy of another being.

Fear

One of the most commonly experienced effects was fear. Some of these were connected to an experience that the self dissolves. As one participant described it: "[I experience fear] because now my self is destroyed. Here and now I disappear." Another participant described experiencing lacking a body, even lacking a sense of space during meditation, and then losing the ability to identify with anything during meditation. Then a voice appeared, "half felt, half heard", telling him that if he wants to proceed, he must become much more radical. As the participant noted, using Anthroposophic terminology, "this is the guardian of the threshold", a kind of being or sensed presence which is perceived both as something outside of oneself as well as part of oneself, representing the aspects of oneself that one has disowned (Steiner 1992). In order to proceed, a kind of death experience occurs at some point, where there is nothing left of the person. The participant's reaction to this was "really great fear", but also wonder and joy. The fear was described further as a "great existential fear, but in the way that … that one can say, yes, this is good." There were many similar experiences of fear appearing in conjunction with a sensed presence. For one participant, such a meeting resulted in a phobia of seeing himself in a mirror (as this gave rise to the same experience of fear as when the sensed presence was there), while another participant described developing agoraphobia after experiencing a threatening sensed presence in the middle of the night, while sleeping. Finally, one participant experienced strong fear connected to an out-of-body experience occurring during mantra meditation, which led to avoiding meditation for years afterwards.

Peace and Calmness

Peace and calmness were regularly cultivated by the Anthroposophic practitioners of the current sample, but also appeared spontaneously as part of practice. Some described a general sense of becoming calmer during meditation yet there was also one description of:

"[ … ] a deep peace, to the point of [ … ] wanting nothing. Nothing anymore [ … ] not to be attached to anything in the future. Which doesn't mean that I won't re-enter, have to re-enter, and that I can want that again, but I'm not so attached to things so crazily anymore, like before, that something [...] absolutely has to happen. I am, I believe, paradoxically, more calm, on the one hand, and, on the other, much more ready to put things on the line, including confrontations with people."

Sometimes experiences of calmness and peace appeared together with other experiences as well, as indicated above, where it was connected with love, joy, and timelessness, though also connected to becoming more focused and centered.

Sadness and Depression

Two participants mentioned recurring depressive tendencies or episodes that perhaps could be connected to meditative practice. However, no one could point to specific links other than that certain meditative experiences, such as a meeting with challenging encounters with sensed presences, can lead to being thrown off balance. Heredity was also pointed to as a factor. One participant stated that people not responding to his suggestions, based on claims of spiritual insight, made him sad. Another pointed out that Anthroposophic practice is not about happiness and contentment. There are continually new challenges. One always strives towards something, towards transforming the world. Nothing is good enough, and hence, as a whole, it does not make life easier.

3.2.5. Sleep

Sleep experiences relate to dreams and wakefulness during the sleep state. There were general reports about either needing less sleep because of meditation, that falling asleep became easier, or that consciousness generally became more wakeful while falling asleep or during sleep. One participant reported developing a sleeping disorder due to experiments with increasing wakefulness during sleep, but this was unconnected to his Anthroposophic practice that started later in life.

Dreams and Lucid Dreams

A few participants reported changes in their dream life, including having archetypal dreams relating to emotional processing. One participant stated that he experienced an increase in the number of lucid dreams due to meditation practice. Another participant noted a similar increase but saw no connection to her practice. One participant claimed to have learned meditation techniques in the dream state while one stated having either premonitions or experiences relating to past lives when dreaming. Two participants stated receiving moral guidance in the dream state.

Sleep Paralysis

The descriptions of sleep paralysis overlap with the descriptions that are commonly reported as: A sense of not being in the body, hearing a hissing sound, sensing an evil presence, strong fear, and the sense of being strangled. Three participants connected sleep paralysis to their meditation practice. One even stated having such an experience during meditation, though the connection did not appear to be strong. Some participants related this experience to meeting the "guardian of the threshold" and the experience generally appearing to be unpleasant.

Lucid Dreamless Sleep

Some participants reported on experiences happening during sleep that were not characterized as dreams. Such experiences included encounters with sensed presences. One participant reported an initiatory experience relating to the personal significance of earthly existence during a deep, dreamless sleep state. Other participants described having intuitions and inspirations related to daily life.

3.2.6. Embodiment

These experiences include everything that is primarily related to the body: Vitality, energetic sensations, energetic centers (chakras), grounding, sense of boundary, and out of body experiences.

Vitality and Health

Many participants mentioned experiencing feeling enlivened, refreshed, and feeling more vital as a result of their meditation practice. Sometimes these reports involved statements that meditation makes one healthier in general; specifically, the subsidiary exercises are said to have such an effect. One participant stated having experienced "circulatory problems" ("Kreislaufstörung") (dizziness, losing contact with the body) that got stronger when meditating. Exceptions were the subsidiary

exercises, which were perceived to have a stabilizing or grounding effect. One participant described having episodes of powerlessness as a result of his practice while another described times when Anthroposophy itself elicited a reaction of nausea and disgust after an intensive period of reading and reflecting on Anthroposophic books and ideas.

### Energies/Forces

Some participants mentioned sensing energy and forces in their bodies that were related to their meditation practice. Certain streams could be sensed as tingling, vibration, rotation, centering and expanding force. Such forces could sometimes be controlled or located outside of the body. Exchanging energy with the environment was also perceived to be possible. Some participants connected this to an experience that the boundary of the body dissolves and/or starts to flow.

### Grounding

This theme relates to being connected to the "ground" or the body. Some participants described becoming more "incarnated", i.e., more connected to their body and experienced an embodiment of the environment in a broader sense. It included subthemes such as dizziness or vertigo, indicating that grounding is lacking. Sometimes a sense of lightness was experienced as positive even though grounding was lacking. Hence, a lack of grounding was not always negative. One participant described a rapture of light when losing the connection to their body accompanied by a stronger sense of perceptual clarity.

### Energetic Centers

One participant reported on energy centers in the body, referring to them as "chakras". These energetic centers or chakras were said to be possible to rotate, which is connected to using them for perception, such as blockages or warmth felt in certain areas. One participant reported meditation leading to a change to the center of their being. This center, which was "normally in the head," moved to their heart. Two participants described working on exiting the body through the crown chakra, which is perceived as being connected to certain challenges such as dizziness or disruption of the body scheme.

### Out of Body Experiences

Out of body experiences overlap in part with sleep paralysis experiences but do not necessarily happen during sleep or in a hypnagogic or hypnopompic phase. Furthermore, sleep paralysis is centered on the experience of an evil presence and other threatening perceptions. Out of body experiences during meditation can be both light and strong. While some described losing control over their physical body, others described feeling like they do not have a body anymore. For one participant, however, it was clear that a sudden external sound would have been heard or noticed. It is also a question for Anthroposophic practitioners whether out of body experiences are something that is worth striving for. Some had avoided it and put an emphasis on the value of earthly existence, while others actively strived for out of body experiences, even though some dangers, like becoming destabilized, ungrounded, etc., are perceived. The notion is that an out of body experience is in more direct contact with a spiritual reality. The tension between seeking an out of body state and working in the world is also a central topic for many practitioners; typically, the emphasis is put on working in the world and increasing embodiment rather than out of body states.

### 3.2.7. Environment

The reports not only concern the inner life of the practitioners, but also how they relate to their environment: How their practice is integrated into daily life and the social world in general, and how

meditation impacts life competency and relationships. Finally, there is the theme of "meaningful connections".

Relationships

These are experiences connected to the dynamic interaction with the external world, including social relationships, the relationship to nature and the cosmos, but also experiences of loneliness. Excluded are perceptual experiences, such as sensed presences as described above.

Anthroposophic meditation was perceived as leading to an experience of harmony and resonance with the external world. One meditator described feeling more connected to physical reality and to other human beings. This sense of connection can also include nature and the cosmos as a whole. Such a sense of connection can bring with it a deeper understanding of the relationship between self and world, and supports the possibility of acting out of what is perceived to come from the periphery of existence. Some practitioners, however, had experienced a sense of isolation. One participant felt "just completely cut off from everything" while another practitioner stated, "I do, actually, now, after all these years, finally feel the sense of community that I was asking [for earlier], but that's a recent thing, I didn't use to feel that [ . . . ] I think I'm only now beginning to be mature enough to understand what that means. I have been on an ego trip I think." The sense of community could indeed arise and the sense of isolation can be perceived to be related to an overly strong focus on oneself. Still, the sense of community described by this practitioner is related to feeling connected to practitioners who speak about their experiences and did not maintain isolation based on the notion that one should not speak about one's meditative experiences.

The relationship one has to other Anthroposophists could also be challenging. Some meditation practitioners see it as one-sided to only read and discuss Steiner's work and rather focus more on their own progress within meditation. To what extent this is a matter of being overly self-centered in a way that leads to isolation cannot be decided here—it is suffice to say that the way in which meditation can practice can affect the relationship to other human beings can be very complex, involving such issues as doubt and courage, social exclusion and recognition, a sense of disharmony, and a feeling of deep connection.

It can be noted that comments on relationship to teachers are scarce. This is in accordance with the Anthroposophic view that having a teacher is not central to practice. However, there are indeed a few positive descriptions of relationships with teachers, where teachers mainly play the role of someone who provided guidance to the meditation practice. Two participants, however, described an experience with a teacher who ended up being "out of balance" and taking part in a dynamic of a split in the group; some following the "old way" and some following the "new way" as defined by the teacher. This was indeed a significant challenge for the participants who had to find a way to work on their own after having ended the relationship with their teacher.

Life Competency

Some participants described an increase of life competency but also related challenges. Some of them described an increase in life competency in general terms, such as becoming more effective at work, experiencing increased awareness for situations in which one can act or intervene beneficially in daily life situations, becoming more capable of making decisions, and taking action so as to live more in accordance with one's insights and values independently of the opinion of others. Especially in crisis situations, meditation could be perceived as support. For example, when it is necessary to engage with people in a way that involves emotional difficulty, meditation can give the necessary detachment.

However, as one participant stated, although Anthroposophy makes life "insanely more adventurous", one can also feel strained or have the sense that one is being attacked during daily life. Another participant noted that meditation can lead to a form of detachment that leads to becoming ineffectual. One participant also noted that Anthroposophic meditation can make life harder and gave the example of a practice in which one cultivates feeling pain when one encounters untruth.

Finally, one participant claimed that Anthroposophic spiritual practice indeed led to an increase in life challenges. The ability to meet those challenges always, in their experience, grew accordingly, so that it was never too much for them.

Integration

Several participants spoke about challenges relating to how they integrate their practice in their daily life, particularly how they deal with the development of new abilities and traits. A common challenge was not being able to speak about experiences since this was generally not welcome within the Anthroposophic movement until recent years. The challenge was one of loneliness and being cut off from the possibility of exchanging with other practitioners. The change towards openness about meditation experiences that recently occurred within the movement was experienced as a relief.

However, being able to perceive more or having a highly sensitized attentional ability as well as having a perceived ability of spiritual observation could also be a challenge. In one case, as indicated by the practitioner, this may have been a factor leading to depression, although mixed in with other strong factors such as overworking and illness. Another practitioner described challenges related to not gaining acceptance for his own perceived spiritual insights in particular, since this insight was perceived to be of high value to others. When such insights were accepted, the experience of integration was positive. One practitioner mentioned a spiritual experience about the nature of his life purpose leading to questioning his daily work setting. In contrast, one participant described feeling more connected to the physical reality and other human beings, and expressed feeling a sense of wonder in relation to the notion that meditation practice can have such an effect.

Meaningful Connections

Some participants experienced meaningful connections occurring between their spiritual practice and daily life events. One example concerns the Anthroposophic teacher's practice of meditating reflectively on a child in one's school class and then having the right idea at the right time the next day relating to this child. Another participant described working in a way where one first seeks a connection with something, a topic or a spiritual being, and then one focuses on the process rather than wanting an immediate result. Initially, nothing will happen, but at a later point it may. For example, one may have an insight, find a book that provides an answer, or get a hint when waking up in the morning. However as one participant stated, one cannot count on things like this to happen. It is also possible, as another participant stated, that one starts to notice an overwhelming amount of meaningful connections, creating a feeling of "complete chaos", especially if the events are negative.

## 4. Discussion

Anthroposophy typically views the human being as consisting of a body, soul and spirit (Steiner 1995). The soul corresponds to the mind or consciousness, and spirit to the fundamental nature of the human being, expressing itself for instance in principled thinking and ethical action. Spirit is also that which connects the human being to the external world and the cosmos at large, which is, in Anthroposophy, inherently of a spiritual nature. Anthroposophic meditation is conceived within this framework. For example, meditation is thought to disentangle the soul and spirit from the body, so that an underlying reality can appear, which can then later lead to a spiritualization of material reality. The way inwards to a spiritual core in meditation is hence seen as part of a process that ultimately goes outwards into mundane reality, based on an experience of the dissolution of the distinction inside and outside in meditative experience.

This study has uncovered a range of themes and experiences that exemplify how this conception of meditation is realized in the life of contemporary practitioners. This includes a change in one's relation to oneself, development of new capacities and perceptions, experience of emotional highs and lows, increase of awareness into the sleep state, and many different perceived effects in one's body and in relation to one's environment.

However, the way Anthroposophic practitioners frame their practice is often quite far removed from current psychological research on meditation. This can, of course, also be said of traditional Buddhist meditation practice. During the last decades, however, different psychological measures have been developed that target potential psychological effects arising from such practice. A similar endeavor has not been undertaken for Anthroposophic meditation. It is also an open question whether Anthroposophic practice would have any effect, for instance, on different mindfulness scales. Although such work can be undertaken in the future, the result that could potentially be most useful coming out of the current study are unique ways of conceptualizing meditation practice, i.e., ways that are not part of, or only to a small degree part of, current psychological research. This will be the main focus in the following discussion.

*Personal development.* Anthroposophic practitioners value individual, personal development, which contrasts with typical ways of conceiving meditation practice, such as Buddhism and other Indian traditions, where the concept of self is seen as problematic and viewed as connected to suffering. As already indicated, there are conceptions of the true self, for instance, in Buddhist traditions, but this true self is not individual. Furthermore, it may be noted that other contemplative traditions such as Christianity (Merton 2003) and Kabbalah (Fishbane 2009) have a more integrative view of the relation between the self and spiritual development. However, if the true self is treated as universal rather than individual, one may see Christian and Mayahana doctrines and compatible (Kennedy n.d., pp. 35–36), though this view contradicts the Anthroposophic view that the true self has an individual aspect. In a way, Anthroposophic practice may also be seen as opposing mindfulness practice. Developing one's character or ability of decisive ethical action requires not accepting things as they are and exerting force to realize something that lies in the future, while mindfulness is typically thought of as "moment-to-moment, non-judgmental awareness, cultivated by paying attention in a specific way, that is, in the present moment, and as non-reactively, as non-judgmentally, and openheartedly as possible" (Kabat-Zinn 2005, p. 108). Note, however, that there is a discussion surrounding how to conceive of mindfulness, particularly in relation to the connection between awareness, emotional reactions and action (Dorjee 2016; Dreyfus 2011). Specific to the Anthroposophic conception of practice is also that development happens in a way that involves de-repression, crises, and a meeting with or realization of a true or higher self. Studies have indeed shown that people tend to believe in the existence of a true self, although this construct is indeed controversial in the field of psychology (Strohminger et al. 2017). Investigating Anthroposophic meditation further can, however, potentially give us insight into what claiming an experience of a true self actually involves.

*Introspection and contemplative inquiry.* Though introspection has for a long time been out of fashion in psychology, there are some signs that it might be making a comeback (Weger and Wagemann 2015; Bitbol and Petitmengin 2013; Giorgi 2010). Anthroposophic practitioners make use of introspection in a setting of contemplative inquiry and, in particular, the way they combine focused attention and open presence to explore topics may represent an effective method of inquiry (Zajonc 2009). Indeed, the insight aspect of meditation, which is traditionally connected to meditation both in the East and West, has up until now hardly been investigated. A central question is whether standard measurements can be used for such research or whether the method of contemplative inquiry is restricted to the contemplative domain.

*Sensed presences.* Sensed presences represent a growing field of study. For example, it has been shown that sensed presences can be induced in a laboratory setting (Braithwaite 2010) and are prevalent in the context of sleep paralysis (Adler 2011). Recently, a sensed presences scale was developed (Barnby and Bell 2017) Anthroposophic practitioners claim to be able to increase their sensitivity to sensed presences and are able to interact with them. This naturally raises many ontological questions, which can also be bracketed. There are indications that human biology supports the perceptions of presences as a kind of unified field of subjectivity, with which one can interact or sometimes represent (Suedfeld and Mocellin 1987; Cheyne 2001). Whether sensed presences actually exist is, to a certain degree, secondary. The human being may on a fundamental level rely on sensing presences when

engaging with the world, whether the sensed presences are human beings, animals, spirits, a book, nature, etc., and how this happens can potentially reveal new aspects of how the human psyche functions. Investigating how Anthroposophic practitioners perceive and interact with sensed presences may provide insight into such functions.

*Sensing subtle aspects of reality.* Similarly, investigating how Anthroposophic practitioners perceive subtle aspects of reality, such as atmospheres—an established concept in esthetics and phenomenology (Böhme 2013; Schmitz 2014)—may provide us with insight into how subliminal processes influence our perceptions and how they can be accessed through meditation practice. Again, no specific ontological presuppositions are required for this kind of investigation. Studies have already shown that meditators are able to increase access to cognitive and volitional processes (Jo et al. 2015; Fox et al. 2012), and this may be true also for the experience of deeper esthetic qualities such as atmospheres.

*Consciousness in the sleep state.* Meditation is traditionally thought of as providing access to the sleep state. Currently, the field of sleep research is expanding, particularly in relation to understanding the structure of experience during the sleep state (Windt et al. 2016; Thompson 2015; Britton et al. 2014). Anthroposophic practitioners understand their practice as accessing the sleep state and can sometimes report on experiences in such states. Their inherent cognitive value could be explored in addition to, for example, how predictive processes are altered during the sleep state.

*Embodied aspects.* Many of the embodied aspects of practice have yet to be investigated. The current study gives a number of examples of which aspects can be explored further: The sense of vitality in the body, energetic sensations, and grounding dimensions of embodiment in relation to meditation practice. This may be related to work within this subfield of meditation research, investigating, for example, the effects of tai chi, kundalini phenomena, and tummo (Kerr et al. 2013; Kerr 2002; Jung 1999; Kozhevnikov et al. 2013).

*The relationship between practice and daily life.* This study gives examples of how Anthroposophic practice interacts with daily life, for example, one's occupation. This is an area that can be explored further. Several studies have already investigated the implications/effects of meditation and mindfulness interventions on for example stress, resilience, and attention in work or study environment (Rees 2011; Rogers 2013). One avenue of research would be to consider whether Anthroposophic meditation has similar effects, while another would be to consider whether there are any specific effects of Anthroposophic practice in relation to the workplace environment and daily life (in general).

*Meditation challenges.* Studying how Anthroposophic practice relates to meditation challenges is also potentially fruitful. Although this study searched specifically for practitioners that have experienced meditation challenges, only one example was found where there was a potential connection between functional impairment and meditation. The challenges met were almost without exception short-lived and did not impact functionality negatively. This could indicate that meditation challenges are rare among Anthroposophic practitioners. However, the total amount meditation hours in the current sample is much lower when compared to other groups, such as the ones investigated by the VCE study (Lindahl et al. 2017), and there seems to be a connection between the amount of practice and the frequency of challenges (Schlösser et al. 2019). More specifically, there is an association between having retreat experience and challenges. In the Anthroposophic community there is no tradition of doing retreats, which rather are often seen as useless and self-centered (Sparby 2017b). Further studies are needed to establish whether or not Anthroposophic practitioners actually experience fewer challenges. If it turns out that the prevalence of challenges is statistically low in comparison to other groups, the reasons for this may be explored, and it could be investigated, for example, whether certain practices such as the subsidiary exercises supports dealing with challenges (Sparby 2018). A further hypothesis that can be tested is whether the reason for lower prevalence is caution on the side of practitioners due to for example knowledge of potential negative effects that are described in Anthroposophic meditation manuals (Steiner 1992).

## 5. Limitations

A limitation in this study is the gender imbalance and that the participants come from European, and predominantly German- or English-speaking countries. Furthermore, this study investigated the experience of meditators at only one point in time. This means that recent experiences are likely to be highlighted and more accurately described while the description of past events are in the background and possibly less accurate. Additionally, because of the limited sample size, the findings might not cover all types and variations of Anthroposophic meditation experiences. Finally, the connection between specific practices and specific effects was not investigated; reported effects may be a result of living in an Anthroposophic environment and using an Anthroposophic framework of understanding to interpret events that may or may not be a result of meditation practice.

## 6. Conclusions and Future Directions

This study has uncovered 32 themes related to Anthroposophic meditation practice, further divided into seven overarching themes. Some of these themes overlap with psychological constructs and other forms of meditation practice while some are unique to Anthroposophic practice. Recommended areas of further studies have been indicated. Beyond this, future studies can focus on any of the fields described above (personal development, introspection and contemplative inquiry, sensed presences, sensing subtle aspects of reality, consciousness in the sleep state, embodied aspects, the relationship between practice and daily life, meditation challenges) and investigate the prevalence of the different perceived effects of Anthroposophic practice. Such inquiry needs to bear in mind that the spiritual perspective of Anthroposophic meditators often cannot be translated directly into psychological constructs. Regardless, it is of interest to investigate whether Anthroposophic meditation has similar effects as for instance mindfulness meditation and other forms of meditation. However, such lines of inquiry may benefit from developing measures that can also include the effects perceived by people practicing Anthroposophic meditation. The present study and related articles may serve such (this) purpose.

**Supplementary Materials:** The following are available online at http://www.mdpi.com/2077-1444/11/6/314/s1, S1. Interview questions, S2. Codebook.

**Funding:** This research was funded by the Mind and Life Institute grant number 2014-Varela-Sparby, Vidarstiftelsen, Rudolf Steinerstiftelsen, Anthroposophische Gesellschaft in Deutschland, Software AG Stiftung.

**Acknowledgments:** Many thanks to everyone who has been involved in this project in different capacities. Willoughy Britton and Jared Lindahl for the original inspiration; Jared Lindahl, Ulrich Ott, and Michael Tremmel for comments on the drafts of the manuscript; Ulrich Ott and Merijn Fagard for validation of the codebook; Kurt Mathisen for checking the final version of the manuscript.

**Conflicts of Interest:** The author declares no conflict of interest.

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
