# Peer review of "Body, Soul, and Spirit: An Explorative Qualitative Study of Anthroposophic Meditation and Spiritual Practice"

_religions, doi:10.3390/rel11060314_

Round 1
Reviewer 1 Report
The manuscript is interesting and generally clear, although the English needs to be edited for tense agreement, repetitive language (increasing attracting more research interest should, for example, be restate as "Anthroposophy is increasingly an object of academic research.")
A minor conceptual point: the ideas of a "higher self" and "second home" discussed in section 3.2.1.2 could be teased out in more detail; more analysis of what these vocabularies might mean would be helpful to the reader. In section 3.2.7.3 it is note clear what "highly sensitive ability of perception" means. What kind of perception? Perception of objects? Or spiritual perception? Emotional or affect perception? Precision is needed in this section.
On pp.22-23, the author could also make more clear or substantiate the claim that standard meditative practice typically wants to dissolve the individual self. Perhaps that is true in the Buddhist tradition, but not necessarily in the Christian tradition (to give one example).
Reviewer 2 Report
This article focuses on a relatively overlooked area of research – Rudolf Steiner and Anthroposophy’s approach to meditation practice. This is a welcome new angle on the increasingly popular subject of meditation which remains dominated by and large by studies of mindfulness alone. However, due to fundamental problems in the structure, presentation, and analysis of this research, I’m afraid to say that this article is far from ready for submission in an international peer-reviewed journal.
At its simplest, there are numerous instances throughout the text where there are slips in clarity. For instance, the paragraph comparing anthroposophy and psychology fails to clarify how the two can in fact be seen as complimentary. After reading the article, the reader is also none the wiser about what the anthroposophic mantra technique actually entails, how it is practiced, and arguably most importantly, how this might differ from other mantra approaches to meditation. In fact, the entire article fails to clarify how anthroposophic approaches to meditation in fact differ from the vast spectrum of other approaches to meditation.
On a more technical level, the authors spends far too long on the methodology section and needlessly wastes time justifying the use of qualitative methods. More worryingly, the latter includes the false claim that unlike quantitative methods, qualitative methods are able to ‘guard against researcher bias’ (p3). On another note, it is not clear why such a small study involving 30 participants justifies a breakdown into percentage shares of key demographic variables. In light of the small sample, a simpler table containing absolute numbers would be more fitting and useful. Similarly, the results including their analysis are impossible to assess objectively without being able to see the original interview protocol and the list of questions that were asked. The author also fails to take note of the wider context in which the participants responded to these questions. In the end, the extensive list of themes is neither clear nor insightful and lacks an overarching purpose. It is too descriptive and lacks any evaluative analysis.
One of the single most jarring methodological issues relates to the fact that that according to anthroposophy, talking about one’s meditative experiences is seen a potentially detrimental to one’s meditative development. This clearly raises significant risks and limitations on the research design which the authors fails to elucidate in more detail.
One cannot help but remain sceptical about the author’s supposed objectivity in this analysis. Too many claims regarding anthroposophy remain implicit, assumed, or unsubstantiated. Regardless of what the author’s personal position may be, they would do well to stick to the principle of ‘show, don’t tell’ of qualitative research, ie showing the qualitative evidence rather than saying what it means. Also, the author frequently cites their own previous work to buttress highly targeted aspects of the current article. This raises serious suspicions about the supposed originality of the data and argument of this article.
Reviewer 3 Report
This manuscripts presents an overview of methods and results of a qualitative study of 30 practitioners of Anthroposophic meditation. On the whole, the texts in the Background, Methods, Results and Discussion sections are appropriate for the subject and study design, well structured, coherent, and easy to follow. Of the 32 subthemes identified in the analysis, all but one (see below, 1.A) are adequately defined and supported by the described findings.
In the research field of Anthroposophic meditation, this study is a seminal piece of work and the present manuscript is set to become a classical article. For research into (any) meditation practice, the study contributes by confirm some earlier findings but also present new themes and subthemes that might be specific for this type of meditation.
Before it can be published, the authors need to clarify or resolve three major issues. Also, there are a large number of textual details and other minor points to attend to.
1. Major issues
A. Positioning of the present manuscript
This paper is said to be the fifth to appear reporting on the same research study. In lines 32-36 the authors state the key topics of each of the previous four papers. They then write (37-38) “These studies will be summarized in the following, providing an introduction to Anthroposophic mediation”. In the abstract (L 5), the present manuscript is said to present “the methodology and results“ from the same study. What does that mean? Is this an overview paper, [only] summarizing previously published work? Or does it bring content not yet published? If the latter is the case, the nature of the new content should be explained (presumably the description of all the 32 subthemes). If this is an overview paper, that is fine – but it should be explicitly stated in the abstract and optimally also in the title.
B. Subtheme 3.2.7.4 „Coincidences“
I think the term “coincidences” is misleading and the use this construct poorly justified.
In the code book this subtheme is defined as the “experience of meaningful coincidences in daily life”, so the key term “coincidence” is not explained. No inclusion or exclusion criteria are stated. According to Table III, experiences coded as “Coincidences“ were reported by four participants.
In the text, Line 759 “Coincidences” are said to having happened “during the day connected to their spiritual practice”. But in the first concrete example, no coincidence but a sequence of two events separated by one night is described: a teacher meditates on a pupil and the right pedagogical idea comes up “the next day”. Furthermore, the sequence of the two events is not interchangeable, the first being an action implicitly understood as a possible cause of the second. In the second example (Lines 762-764 or 765, possibly the identical case is also cited in the codebook) the distance between the two events is probably longer: “Initially nothing will happen, but at a later point it may… get a hint when waking up in the morning” – So these two examples are no coincidences – they were not concurrent or, citing Jung: “synchronicity” events. The third example (L 766-767) is clearly a coincidence (many coincidences occur at the same time) but the impact on the practitioner is of a totally different kind: instead of new insights, the practitioner experiences a feeling of “complete chaos”.
In conclusion, for the reported second and third practitioner experiences cited among four practitioners, the term “coincidences” does not fit with the nature of the events. “Meaningful sequences” or similar might be used. Also it is hard to see any valid conceptual link between these two reports and the third: many coincidences occurring at the same time, creating a feeling of complete chaos would point in a different, more disturbing direction. Perhaps they could be seen as related to (though not identical with) “self-dissolution” as described in the codebook?
I think the authors should consider changing the words denoting this term and possibly also reclassifying the four experiences. In case they are unable or unwilling to do so, at least they need to discuss the issues raised here adequately.
C. “Body, soul and spirit” and correspondences to the themes identified in the study
The three terms “body, soul and spirit” from the title are introduced in Line 106, citing Steiner’s book “Theosophy”, GA 9. In lines 769-774, the authors briefly present these three terms. In lines 783-787 they write “The themes covered here may be seen to correspond roughly to the traditional Anthroposophic themes in the following way: Body corresponds to embodiment, soul to affect and cognition, and spirit to self. Hence the themes environment, sleep, and perception have no direct correspondence to the traditional threefold model of Anthroposophic anthropology.” I’m afraid I think this description is totally inadequate. My best advice is to omit it altogether in this paper, also because this type of correspondence seeking is no central topic of the paper.
- The choice of anthropological “models” within Anthroposophy is too narrow (cf. e.g. further differentiation into to sevenfold and nine-fold models in the source cited, GA 9; and the classical fourfold typology described in GA 27).
- Correspondences of themes identified in the study to Anthroposophic anthropology should not be limited to one anthropological entity within a model but also include the interactions between several entities.
- For example, ordinary sense perception involves the soul (~”sentient organization” in the fourfold conception) and spirit (~ “I” or “I-organization”), and in sleep, these two are partly separated from the body (i.e. from the physical and life organizations of the fourfold model). Among the subthemes of the “environment” theme, “Life competency” and “Integration” could be seen as representing a further development of the human “spirit” according the threefold model.
- Hence, the statement that “the themes environment, sleep, and perception have no direct correspondence to the traditional threefold model” is misleading, because it can lead the reader to believe that these themes have no counterparts in the anthroposophic conception of the human being, which is clearly not the case.
2. Minor issues
|
Line |
Subject/words |
Comments and/or suggestions |
|
22-23 |
A recent milestone has been the publication of the critical edition of Steiner’s work (Steiner 2013) |
A recent milestone has been the inauguration of a critical edition of Steiner’s written work (Steiner 2013) |
|
25-26 |
If we include Anthroposophic medicine (Kienle et al. 2011; 25 Harald Johan Hamre et al. 2014; Scheffer et al. 2012; Harald J. Hamre et al. 2007) |
The authors cite one HTA report update (Kienle 2011), two reports of results of the AMOS study (Hamre 2012 + 2007) and a paper on education in A. medicine in an academic centre. I think an overview presentation of A. medicine should come first, the most recent comprehensive is still this: Kienle GS, Albonico H-U, Baars E, Hamre HJ, Zimmermann P, Kiene H. Anthroposophic Medicine: an integrative medical system originating in Europe. Global Adv Health Med 2013;2(6):20-31. The rest is discretionary. But if you want to cite AMOS, Hamre 2007 can be omitted, Hamre 2014 is sufficient. |
|
32-37 |
Four analysis have been published from this research project… |
This very important information is not well placed within the text. Currently it interrupts the narrative on Anthroposophic meditation. Thus it can be overlooked. Therefore I suggest you shift these sentences to the end of the Introduction section (currently line 98 onwards) and merge them with the text there. Also, the text should be followed by a clear positioning of the present paper (cf. 1.A, above). |
|
37 |
6x „mediation“ (also in lines 56, 201, 617, 632, 697) |
6x meditation |
|
40-41 |
the theosophical movement, which have brought many of the Asian concepts and ideas |
“Asian” is very nonspecific. I suggest you change to “the theosophical literature by Helena P. Blavatsky and others”. As Steiner translated verses by Blavatsky and Mabel Collins for meditation purposes, that would be much closer to the topic of your paper. |
|
42-51 |
Anthroposophic meditation can be seen… |
A nice, relaxed narrative description. Perhaps a brief statement on any features demarcating this type of practice from other meditation schools could be worthwhile here. |
|
48 |
based in spiritual insight |
based on spiritual insight |
|
55-56 |
The secondary literature on Anthroposophic mediation… |
I think also the primary literature on anthroposophic meditation should be explicitly in the introduction: Steiner’s seminal instruction book GA 10 (Steiner 1992, already cited in line 884) and the old GA 245 edition, which contains major texts on meditation in Steiner’s lectures. |
|
64, 67 |
2x Anthroposophic “doctrine” |
“Doctrine” sounds value-laden and, I think, does not do justice to the subject. Unless this is an established term in the research field of the paper, it should be replaced by “conception”, “paradigm” or similar. |
|
74, 80 |
complimentary |
„complementary“ is what you mean here |
|
78-79 |
there is nothing in psychology that makes it inherently reductive. |
Discretionary: I suggest this rather sharp formulation is re-stated, e. g: “psychology research need not necessarily be reductive” |
|
81-97 |
[You present results from the present study] |
This content should come in the section 3. Results. |
|
107 |
Anthroposophy(Steiner 1995). |
Anthroposophy (Steiner 1995). |
|
140 |
The research design of this study was inspired by the methodology… |
Replace “inspired” with “adapted from”, “informed by” or similar. |
|
151-167 |
[Discussion of possible bias] |
Discretionary: This text could be shifted to discussion section. Or, in case it is covered in your publication “discussing methodological issues“ (L 33) it could be shortened or omitted. |
|
152-3 |
the interviewees …has |
have OR had |
|
177-8 |
a participant … changed their mind |
his OR her |
|
190 |
process of liberalization |
process towards openness [about meditation experiences] |
|
234 |
[As lines 177-8] their |
his OR her |
|
Table I |
range: 1,5-41 |
range: 1.5-41 |
|
270 |
Practices Numbers |
Practices. Numbers |
|
Table II |
The daily review: |
The daily review (“Rückschau”): |
|
Tables II-III |
Inconsistent use of parenthesis and colon, e.g. p.8: Biographical work (1 (3%)) / The calendar of the soul/poems: 3 (10%) |
Please regularize: Biographical work: 1 (3%) The calendar of the soul/poems: 3 (10%) |
|
289 |
A more in-depth analysis may be presented in a further article. |
Are you sure this will be published? If yes, replace “may” with “will”. If no, I advise you to delete this statement. |
|
290 |
Prevalance |
Prevalence |
|
290 |
Overview of Themes and Prevalance. Numbers indicate prevalence in the present sample (N=30). |
Prevalence of Themes in the study sample (N=30). |
|
Table III |
|
How was the sequence of sub-themes determined? Unless you have conceptual or other content-related reasons for a specific order, a natural sequence would be the alphabetical order of the first word of each sub-theme. |
|
297-8 |
development from the formed to the latter |
development from the former to the latter |
|
344-767 |
[use of past and present tense when summarizing practitioners’ narratives] |
In these summaries (not the verbatim citations) of the practitioners narrative, the past tense (“he had”… “she did”) is mostly used – but in a relevant proportion you shift to the present tense (“he has”), sometimes even within a sentence. I strongly suggest you put all this in the past tense and reserve the present tense for your own comments in these texts (e.g. as you do in line 365: “This experience is an example of…”). That will make it easier for the reader to immediately recognize if the content comes from the practitioners or the authors. |
|
352 |
“threshold experience”, …the old has been left behind |
Perhaps this is more appropriate? “threshold experience”, …the old is getting left behind |
|
357 |
on occasion reported on experiences |
occasionally reported on experiences |
|
370 |
the following sub-themes: Thinking, knowledge, memory, sense of reality, and conviction, |
The sequence of subthemes is different from the following headings 3.2.2.1-3.2.2.4. Also the term “sense of reality” has not heading. Please regularize. |
|
423 |
were motivation |
was motivation |
|
430 |
And that is what I mean, that meditation … |
I suggest that these verbatim citations from the interview transcripts are marked with quotation marks “…”, and possibly also set in italics. |
|
439 |
meta-cognition, i.e. the ability to be… |
It would be helpful for some readers if you shift this definition of meta-cognition to the beginning of this section (Line 426). |
|
447 |
useful for having new content appear |
useful for enabling new content to appear |
|
466, 470 |
2x angle |
2x angel |
|
492-3 |
there are many different qualities that are |
many different qualities are |
|
496 |
it will not be possible |
it is not possible |
|
501 |
images had contained |
images contained |
|
502-3 |
and other images were experienced as healing or invigorating, though they could also be cold. |
Does “could also be cold” refer exclusively to the images that “were experienced as healing” or does the term also refer to the other types of images with “an auditory or cognitive component” (lines 501-2)? |
|
517 |
somatic character |
(Suggestion): bodily character |
|
481-2 |
The traditional term for visual experiences in Anthroposophy is “imagination” |
“Inspiration” in the corresponding sense recurs in lines 526, 527, 639; “intuitions” at 639. Altogether I think it is warranted to briefly describe these three terms as they are used in anthroposophy in the context of meditative practice, citing Steiner GA 12 “Die Stufen der höheren Erkenntnis” as the classical sourcebook. |
|
530-1 |
something that is working its way into one’s thinking at representational capacity |
What does “at representational capacity” mean here? |
|
582 |
“the …guardian of the threshold”, a kind of being… which is perceived … representing the aspects of oneself that one has disowned |
Where did this description come from? If it was the practitioner, you must write “was perceived”. If this is the authors’ description, you could add a citation of Steiner GA 9 (already cited as Steiner 1992), also since the term recurs at Line 633. |
|
603 |
peaceappear |
peace appear |
|
610 |
Heritage |
Heredity |
|
648-9 |
experienced circulatory problems (dizziness, losing contact with the body) that got stronger when meditating |
From a medical point of view, “circulatory problems” does not make sense here. According to the symptoms, “circulatory problems” would be postural hypotension. In that condition, the symptoms would appear or increase strongly when standing up, not when meditating (presumably while sitting). From the context and the cited symptoms, “dissociative symptoms” appears much more appropriate. (In case the practitioner was German-speaking, the Term “Kreislaufstörung” is sometimes used in a context of symptoms popularly but wrongly believed to arise from the circulatory system.) If you cannot change the wording, you must write “circulatory problems” with quotation marks. |
|
661-3 |
connected to the “ground“ … “incarnated” subthemes such as becoming dizzy |
How is “dizziness” to be understood as a subtheme of “connection to the ground“ or becoming “incarnated”? Please explain and justify this classification, or alternatively, reconsider it. |
|
664 |
Hence a lack of grounding is not necessarily negative. |
This also seems tenuous to me, cf. above, lines 661-3. |
|
700 |
…can bring with it a deeper understanding of the relationship and that it can become possible… |
Awkward formulation, please check and revise. |
|
712-714 |
For some… seen as the most important ways… [others] see it as one-sided. To what extent this is a matter of being “on an ego trip” cannot be decided here |
The phrase “a matter of being “on an ego trip”” is unclear to me. To whom of the two groups does it refer? Also, I think the term “ego trip” is too colloquial here. |
|
719 |
the teacher is not central |
Perhaps rather: “having a teacher is not central”? |
|
730 |
taking action top live |
What does “top live” mean? |
|
741 |
never was |
was never |
|
747 |
The change |
The change towards openness about meditation experiences |
|
750 |
this may have been a factor leading to depression |
Is this the opinion of the practitioner or the authors? |
|
756 |
even wondered |
even wonder (?) |
|
774 |
meditation is thought to disentangle the soul from the body |
Is there no disentanglement of the spirit? Can you give a reference for this statement? |
|
836 |
the human psyche functions |
how the human psyche functions (?) OR the functions of the human psyche (?) |
|
853 |
many aspects that embody practice that have yet to be investigated |
Unclear phrase. Please reformulate. |
|
872-3 |
the amount of practice in the current sample |
Do you mean “duration of regular practice ([in] years)” as in Table I? Or the amount of time spent each day with meditation? Please specify. |
|
877 |
no tradition of doing retreats, but rather are often seen |
“no tradition of doing retreats, which rather are often seen” (?) Please clarify this phrase. |
|
880 |
it can be investigated |
it could be investigated |
|
882 |
less prevalence |
lower prevalence |
|
890-1 |
the limited sample size could mean that the findings merely reflect the experiences of the participants involved |
This statement is not very informative. Perhaps the following, or similar, might reflect what you want to say? “because of the limited sample size, the findings might not cover all types of experiences of practitioners of anthroposophic meditation” |
|
1017, 1020 |
Steiner 1975 = English translation of Steiner 1995 |
Please eliminate this redundancy. When citing Steiner editions (except the SKA in Line 1023), there should be a consistent approach to cite either the German original source or an English translation or both. |
|
1021 |
2013. Schriften. Kritische Ausgabe |
EITHER: “2013-. Schriften. Kritische Ausgabe“ indicating the ongoing project OR cite the full title of the book that was published in 2013. |
Round 2
Reviewer 3 Report
All my major and minor concerns in my previous review have been adequately met. Thus, the manuscript has been significantly improved and now warrants publication in Religions.
Before publication, three minor text amendments should be performed (by the authors or typesetters).
|
Line |
Words |
Suggested amendment |
|
21 |
range experiences |
range of experiences |
|
67 |
datarather |
data rather |
|
74 |
complimentary |
complementary |